# Ferroptosis: Cancer Stem Cells Rely on Iron until “to Die for” It

**DOI:** 10.3390/cells10112981

**Published:** 2021-11-02

**Authors:** Emma Cosialls, Rima El Hage, Leïla Dos Santos, Chang Gong, Maryam Mehrpour, Ahmed Hamaï

**Affiliations:** 1Institut Necker-Enfants Malades (INEM), Inserm U1151-CNRS UMR 8253, Université Paris Descartes-Sorbonne Paris Cité, F-75993 Paris, France; emma.cosialls@inserm.fr (E.C.); rima.elhage@inserm.fr (R.E.H.); leila.dos-santos@inserm.fr (L.D.S.); 2Breast Tumor Center, Sun Yat-sen Memorial Hospital, Guangzhou 510120, China; gchang@mail.sysu.edu.cn

**Keywords:** iron metabolism, ferroptosis, autophagy, cancer stem cells

## Abstract

Cancer stem cells (CSCs) are a distinct subpopulation of tumor cells with stem cell-like features. Able to initiate and sustain tumor growth and mostly resistant to anti-cancer therapies, they are thought responsible for tumor recurrence and metastasis. Recent accumulated evidence supports that iron metabolism with the recent discovery of ferroptosis constitutes a promising new lead in the field of anti-CSC therapeutic strategies. Indeed, iron uptake, efflux, storage and regulation pathways are all over-engaged in the tumor microenvironment suggesting that the reprogramming of iron metabolism is a crucial occurrence in tumor cell survival. In particular, recent studies have highlighted the importance of iron metabolism in the maintenance of CSCs. Furthermore, the high concentration of iron found in CSCs, as compared to non-CSCs, underlines their iron addiction. In line with this, if iron is an essential macronutrient that is nevertheless highly reactive, it represents their Achilles’ heel by inducing ferroptosis cell death and therefore providing opportunities to target CSCs. In this review, we first summarize our current understanding of iron metabolism and its regulation in CSCs. Then, we provide an overview of the current knowledge of ferroptosis and discuss the role of autophagy in the (regulation of) ferroptotic pathways. Finally, we discuss the potential therapeutic strategies that could be used for inducing ferroptosis in CSCs to treat cancer.

## 1. Introduction

Iron is an essential nutrient in all mammals and is involved in key biological processes (as a catalytic component of various proteins), including hemoglobin synthesis (heme), DNA synthesis (ribonucleotide reductase), oxygen transport (hemoglobin), mitochondrial respiration (electron transport chain), energy metabolism (aconitase, succinate dehydrogenase), detoxification (cytochrome P450 enzymes), antioxidant defense (catalase), oxygen sensing (hypoxia-inducible factor (HIF) and prolylhydroxylases) and immune defense (myeloperoxidase). This property is based on the chemical transitional ability of iron to fluctuate between an oxidized form (Fe^3+^, ferric state, electron acceptor) and a reduced form (Fe^2+^, ferrous state, electron donator) in various enzymatic or redox reactions. However, although iron is tightly regulated, excess free iron in cells can also contribute to the formation of free radicals from reactions with oxygen and excess free radicals, leading to lipid peroxidation, the increased production of reactive oxygen species (ROS), oxidative stress, and DNA damage. Thus, iron represents a double-edged sword. Indeed, the accumulation of iron and ROS is linked to various pathologies, including iron overload diseases and cancer. Furthermore, cancer cells exhibit increased iron demand compared to non-cancer cells. In line with this, the pathways of iron uptake, storage, mobilization, trafficking, and regulation are all perturbed in cancer, suggesting that the reprogramming of iron metabolism is a central aspect of tumor cell survival. Anemia is frequently observed in many patients with cancer, and iron homeostasis dysregulation is implicated in numerous types of cancers [1,2,3]. The results of several experimental and epidemiological studies support the effect of dietary and systemic iron on tumor development. Readers interested in more details about iron homeostasis and disorders in cancer cells and tissues should read the several recent reviews on this topic [4,5,6,7]. Importantly, recent studies have shed light on the role of iron metabolism in cancer stem cells (CSCs) and suggested that the specific targeting of iron metabolism in CSCs may improve the efficacy of cancer therapy. This iron dependency can make CSC and non-CSC cells more vulnerable to a newly identified form of programmed cell death, referred to as ferroptosis. This cell death process characterized by the iron-dependent accumulation of lipid peroxides is morphologically, biochemically, and genetically distinct from other well-known forms of regulated cell death, including apoptosis, various forms of necrosis, and autophagy. In some cases, metabolic reprogramming has been linked to an acquired sensitivity to ferroptosis, thus opening new opportunities to treat tumors that are unresponsive to other conventional therapies.

## 2. Iron and CSC

Characterized by several markers (CD44, CD24, ALDH1, and CD133 which are summarized in Table 1; for a review, please see [8]), CSCs are subpopulations of cancer cells within liquid and solid tumors that share similar features to those of normal progenitor/stem cells, such as self-renewal and multi-lineage differentiation abilities, which drive the tumor growth and heterogeneity. Demonstrated to be resistant to conventional therapies both in vitro and in vivo, CSCs are thought to be responsible for tumor recurrence and metastasis [9]. Under the CSC paradigm, all tumor cells are not uniform, but tumors fit in a hierarchical organization driven by CSCs [9]. A new relationship between CSCs and iron has recently been highlighted by several laboratories including our team (now called FEROSTEM: “FER” for iron and “STEM” for stem cells).

### 2.1. Iron Homeostasis at Cellular Level

Briefly, the transferrin (Tf) pathway is mainly used by both normal and cancer epithelial cells for iron uptake. The complex formed between Tf associated with 2 ferric ions (Fe^3+^) binds to its receptor transferrin receptor 1 (TFR1/TRFC), at the cell surface and is endocytosed. In endosomes, these ions are released at low pH (5.2–5.5) and reduced to Fe^2+^ by the ferrireductase six epithelial transmembrane antigens of the prostate 3 (STEAP3). Once in their ferrous form (Fe^2+^), they are transported to the cytosol by divalent metal ion transporter 1 (DMT1/Nramp2). The Tf/TFR1 complex is then recycled to the cell surface. Other iron/metal transporters at the membrane surface have been described, including ZIP8/14, which functions optimally at pH 7.5 in the non-transferrin-bound iron pathway. Once in the cytosol, Fe^2+^ ions constitute a labile intermediate pool or labile iron pool (LIP). Cellular free iron participates in several biological processes in different cellular compartments, as described above. As excess amounts of free iron can be toxic for the cells, it is stored in ferritin or exported by ferroportin (FPN) with the assistance of the ceruloplasmin ferroxidase [4]. Ferritin is a macromolecular complex with 24 subunits formed by light and heavy chains of ferritin (FTL and FTH, respectively) with the ability to store up to 4500 iron atoms. Iron exportation is regulated at the systemic level by hepcidin (a peptide hormone secreted by the liver and a master regulator of systemic iron metabolism) by binding FPN and promoting its phosphorylation and subsequent lysosomal degradation [36,37].

### 2.2. Iron Metabolism Dysregulation as a Hallmark of CSCs

In the case of CSCs in glioblastoma, Schonberg et al. reported an increase in the expression level of transferrin and TFR1 compared to non-CSCs [38]. They demonstrated that these CSCs uptake iron from the microenvironment more efficiently than their non-CSC counterparts through iron-tracing experiments. At the functional level, the authors showed that TFR1 and ferritin are crucial for the maintenance of CSCs in vivo underlining the crucial role of iron in these subpopulations. In breast cancer, we showed that cellular iron and Tf uptake, which is correlated with a higher level of TFR1 expression, is more robust in CSCs than in non-CSC counterparts [39,40]. Overall, this finding supports the existence of enhanced iron trafficking in CSCs, underlining the importance of iron in the behavior of these subpopulations. Moreover, the first demonstration of the novel role of iron via hydroxyl radicals in CSC regulation was performed in non-small lung cancer cells, showing its importance in aggressive cancer behaviors and likely metastasis through SOX9 upregulation [41]. In other in vitro CSC model based on tumorspheres derived from MCF-7 cells, iron uptake, LIP, iron mitochondria, and cell death induced by iron chelators are also enhanced in spheres compared to monolayer cell cultures [42]. Furthermore, the authors identified a transcriptional signature based on 10 genes related to iron metabolism to distinguish MCF-7 cells resistant to tamoxifen that display CSC features, as well as to distinguish CSC leukemia from non-CSC leukemia in a mice model of acute promyelocytic leukemia. Accordingly, FPN is also down-regulated in cholangiocarcinoma CSCs, promoting iron retention. In addition, low FTH levels and high TFR1 expression (as a typical pattern indicative of low iron needs) are displayed in cholangiocarcinoma cells grown in monolayers, whereas the opposite situation occurred when the same cholangiocarcinoma cell lines were allowed to form tumorspheres, accompanied with an increase in iron content and oxidative stress [43]. In line with this, through GDF (growth differentiation factor)15/SMADs-mediated regulation, hepcidin has been shown to be upregulated in MCF-7 spheroid cell cultures compared to 2D-monolayer conditions [44]. GDF15/MIC-1, which is a member of the TGF-b superfamily [45], has been even described to play a role in the enhanced invasion as well as maintenance of breast CSCs [46,47]. In ovarian cancer, an increase has also been seen in the expression level of TFR1, in contrast to a decrease in the expression level of FPN in CSCs compared to their non-CSC counterparts. This promotes higher levels of intracellular iron and hence a higher dependency on iron in CSCs. This is supported by the fact that intracellular iron reduction interferes with their proliferation in vitro and their tumorigenicity in vivo. In addition, iron increases metastatic spread by facilitating invasion through the expression of matrix metalloproteases and IL-6 synthesis [48]. Accordingly, iron supplementation has been found to promote CSC-like features in breast [39,40], lung [41], and cholangiocarcinoma cancer cells [43], whereas exposure to iron chelator was shown to have an opposite effect on spheroid formation ability in lung [41] and cholangiocarcinoma cancer cells [43]. Overall, this supports the notion that CSCs in various histological tumors are heavily dependent on iron; however, the role of iron in the biology/plasticity of CSCs remains to be clarified.

### 2.3. Iron-Related Stemness Features/Markers

As illustrated in Table 1, some CSC-related markers or signaling pathways have been described to be connected with iron metabolism. Figure 1 summarizes the role of iron metabolism in supporting CSC maintenance in a comparison of cancer cells. For example, CD44, which is frequently used as a CSC marker, has been demonstrated as an actor in iron acquisition in CSC by the glycosaminoglycan-mediated endocytosis of iron through its interaction with hyaluronates [10]. In addition, CD44 itself is transcriptionally regulated by nuclear iron illustrating a positive feedback loop in contrast to the IRP-mediated negative regulation of TFR1/DMT1 by excess iron. CD133 is also itself regulated by iron and its expression at the plasma membrane has an impact on the endocytosis of Tf/TFR1 or iron uptake supporting the existence of a transferrin–CD133–iron network [18]. Moreover, iron is able to increase the expression of CSC markers through the WNT signaling pathway. Indeed, a high-throughput WNT inhibitor screen revealed the critical iron dependance of beta-catenin/WNT signaling in cancers and iron chelation represents as an effective way to inhibit WNT signaling [49]. In line with this, iron exposure is also thought to affect the expression of Hedgehog Interacting Protein Like-2 (HHIPL2), an inhibitor of the hedgehog signaling pathway [50], and, contrastingly, to increase the expression of GLI1, promoting the self-renewal and maintenance of CSCs by activating the transcription of stemness genes such as *CD44* and pluripotent transcription factors including *NANOG*, *SOX2* and *OCT4* [51].

### 2.4. Iron Regulation and Stemness Behaviors

Through the JAK/STAT3 pathway, which has been demonstrated to be an important pathway for CSC plasticity, both interleukin(IL)-6 and oncostatin M (OSM) (belonging to the same family of pro-inflammatory cytokines) are robust inductors of CSCs or promote the selection and expansion of CSC subpopulations [52,53,54,55]. Notably, we demonstrated that concomitant with the emergence of breast CSCs, OSM increases the expression of ferritin and the down-regulation of ferritin expression by RNA interference affects the OSM-induced enrichment of CSCs, confirming the crucial role of iron metabolism in the maintenance/plasticity of CSCs [39]. In addition, the silencing of FTH expression is also able to modulate the expression of some CSC markers and spheroid formation in other solid tumor types [38,56]. Several studies have supported the notion that redox status has an important impact on stem cell maintenance [57,58]. Indeed, high levels of ROS lead to both the loss of self-renewal and differentiation and the enhancement of the radio-susceptibility of CSCs in several types of cancer, such as breast, glioblastoma and prostate [59,60,61]. Interestingly, FTH with its ferroxidase activity represents a major antioxidant protein limiting iron-mediated oxidative stress. Accordingly, it has been shown that the *FTH1* gene silencing in human embryonic stem cells (ESCs) leads to the overactivation of the nuclear factor (erythroid-derived-2)-like 2 (NRF2) signaling pathway and pentose phosphate metabolic pathway (PPP) to maintain the redox status [62]. Nevertheless, the silencing of FTH1 in human erythroleukemia blast cells affects their erythroid fate underlining the role of FTH in cancer cell differentiation [63]. In addition, the downregulation of FTH expression overcomes chemoresistance in solid tumors through the modulation of ROS [64]. With regard to CSC expansion, it is well established that the epithelial-to-mesenchymal transition (EMT) process in cancer cells is accompanied by the acquisition of CSC properties [65]. In line with this, FTH expression is able to modulate both the EMT in a variety of solid cancer in vitro models [66,67,68] and the EMT-like trans-differentiation process in hematological cancer models [69], mainly (but not exclusively) through its ability to regulate the amount of iron-dependent ROS. Accordingly, the depletion of ferritin in glioblastoma stem-like cells affected their proliferation/cell cycle progression through the STAT3-Forkhead box protein M1(FOXM1) regulatory axis, revealing an iron-regulated CSC pathway [38]. STAT3 activation and the transcription factor FOXM1, which is both downstream target gene and inductor of STAT3, thus constituting an activation feedback loop, are required to promote glioblastoma CSC self-renewal and tumorigenicity [38,70]. In particular, the authors postulated that the increased level of ferritin in CSCs directly interacts with STAT3 and/or potentiates STAT3 phosphorylation and leads to the activation of downstream signaling targets including FOXM1, thus illustrating iron metabolism via a ferritin—STAT3—FOXM1 feedback loop. In line with this, AlkB homologue 5 (ALKBH5), a demethylase of the mRNA modification N6-methyladenosine (m^6^A), regulates the expression of the *FOXM1* gene by acting in concert with a long noncoding RNA antisense to *FOXM1* (FOXM1-AS) on pre-mRNA stability in glioblastoma CSCs [71]. ALKBH5 belongs to the AlkB family of nonheme Fe(II)/alpha-ketoglutarate-dependent dioxygenases which are essential regulators of RNA epigenetics (also called Epitranscritomics), and thus regulate gene expression and cell fate [72]. ALKBH5 also has *NANOG* mRNA as a target in breast CSCs [71]. Furthermore, ZNF217, an m^6^A methyl-transferase inhibitor, inhibits the m^6^A modification of several pluripotency factor mRNAs, including *NANOG*, *KLF4* and *SOX2* in breast cancer to promote the CSC-like phenotype and breast cancer metastasis under hypoxic conditions [73]. On other hand, ZNF217 acts as a transcriptional repressor that inhibits FPN expression, leading to intracellular iron retention, increased iron-related cellular activities, and enhanced prostate cancer cell growth [74]. Mechanistic investigations have demonstrated that ZNF217 facilitates the H3K27me3 levels of *FPN* promoter by cooperating with the histone methyltransferase EZH2 to suppress FPN expression. The hypermethylation of the *FPN* promoter is also associated with a decreased FPN level in breast cancers [75]. Other iron-related genes are also subject to specific epigenetic modifications. The level of H3K4 methylation, which is associated with a transcriptionally active form, in the *HAMP* gene promoter coding hepcidin is specifically increased under exposure of BMP-4 or TGF-b [76]. The FPN/hepcidin axis could be a major iron-mediated node for controlling cancer in particular CSCs. In addition, the inhibitors of histone deacetylases (HDAC) enzymes that remove acetyl groups from histones leading to a condensed chromatin state and transcriptional repression, induce FTH expression through the recruitment of NF-Y transcription factor to the *FTH* promoter [77]. G9a, an H3K9 methyltransferase associated with HDAC1 and YY1, a member of the Krüppel family of transcription factors, forms a silencing multi-molecular complex targeting the repression of the ferroxidase *Hephaestin* gene, which codes a co-factor of FPN involved in ferric export in breast cancer. Furthermore, a tissue microarray analysis from 75 breast cancer patients revelated that high G9a expression and low hephaestin expression are associated with poor prognosis [78]. Interestingly, iron-chelating agents including deferoxamine (DFO), deferasirox (DFX), and their synthetic derivatives inhibit epigenetic JumonjiC domain-containing histone lysine demethylases (JmjC KDMs) which are Fe(II)/2-oxoglutarate-dependent oxygenases that are also involved in transcriptional regulation and DNA repair [79,80]. The first demonstration has been performed by Cao et al., showing that DFO induced a significant increase in global histone methylation in colorectal cancer, leading to the dysregulation of many cell growth-related genes [81]. Iron chelation impairs the enzymatic activity of KDM2B, KDM3B, and KDM4C, affecting the demethylation of H3K9 and cyclin E1 expression. It was postulated that CSCs arise through epigenetic changes [82]. In particular, specific to the KDM5 family H3K4 histone lysine demethylases, KDM5A (named JARID1A/RPB2) was identified to be overexpressed in drug-resistant cells displaying some CSC-like features [83], and the overexpression of the KDM5B (named JARID1B/PLU1) is a marker for identifying a subpopulation of human melanoma CSC-like cells [84]. Altogether, these findings support the existence of an iron-mediated network/feedback loop that mediates the regulation of both CSC-related pathways and epigenetic programs and that is a potential target for novel therapeutic strategies against CSC subpopulations.

## 3. Ferroptosis, Iron-Driven Cell Death

### 3.1. Iron Accumulation and Lipid Peroxidation: Drivers of Ferroptosis Execution

Recently, a new type of iron-dependent programmed cell death has been described and named ferroptosis [85]. Many inductors (including erastin, RAS-selective lethal molecule 3 (RSL-3) or pharmacological/clinical drugs such as sorafenib, sulfasalazine, artesunate) were identified before the concept of ferroptosis emerged [86]. Since, the specific inhibitors have been identified, such as liproxstatin-1 (Lip-1), ferrostatin-1 (Fer-1), and vitamin E (VitE), or coenzyme Q10 (CoQ10), as well as their analogs acting as (lipid) ROS scavengers (for reviews, please see [86,87]). Indeed, ferroptosis is morphologically, biochemically, and genetically distinct from other well-known forms of regulated cell death, including apoptosis (caspase-dependent), various forms of necrosis (RIPK1&3/MLKL dependent), and autophagy (ATGs-dependent) (for recommendations of the nomenclature on cell deaths, please see the review [88]). Morphologically, cells undergoing ferroptosis experience a reduction in cell volume, with intact cell membrane devoid of blebbing; they are lacking chromatin condensation, and have increased mitochondrial membrane density with vestigial cristae and outer mitochondrial membrane rupture [85,89,90]. Biochemically, ferroptosis is the result of excessive iron-dependent lipid peroxidation (LOOH) from oxidized polyunsaturated fatty acids (PUFAs)-containing membrane phospholipids. It leads to large molecular damage on proteins, nucleic acids, and lipids [85,91]. Among the phospholipids that are oxidized during cell death, arachidonic acid (AA)- or adrenic acid (AdA)-containing diacylated phosphatidylethanolamines (PE) have recently been identified as ferroptotic death signals/precursors by genetic, bioinformatics, and LC-MS/MS lipodomics approaches [92]. Acting as lipid death signals or as the direct executioners of ferroptosis, reactive lipid derivatives can directly or indirectly promote cell death by binding covalently to essential intracellular proteins and thus inactivating them [93]. Oxidizable PUFAS are crucial for the execution of ferroptosis. Thus, the genetic and/or pharmacological inhibition of their incorporation into the cellular membrane, by acyl-CoA synthetase long chain family member 4 (ACSL4) and lysophosphatidylcholine acyltransferase 3 (LPCAT3), or the inhibition of the oxidation of PE-esterified AA and AdA by 15-lipoxygenase (15-LOX, inhibited directly by VitE) protects cells against ferroptosis [94]. In addition to being essential in the enzymatic oxygenation reaction of PUFAs (for example, 15-LOX, an iron-binding enzyme) via the Fenton chemical reaction with hydrogen peroxide (H_2_O_2_), excessive level of ferrous ions (from transferrin, ferritin, or again heme) fuels electron-driven lipid peroxidation by the production of hydroxide (OH^−^) and hydroxyl radicals (·OH). Iron level regulation is the key actor in ferroptosis execution, as demonstrated by its uptake increased and by the inhibitor action of iron chelators (such deferoxamine (DFO), desferrioxamine mesylate (DFX), and ciclopirox olamine) [95]. Moreover, the addition of exogenous iron (e.g., ferric ammonium citrate, ferric citrate, and iron chloridehexahydrate) sensitizes cells to ferroptosis [85]. In line with this, (Holo-)transferrin (loading for ferric ions Fe^3+^) was also identified as an essential regulator of ferroptosis [96] and transferrin receptor 1 (TFR1/TFRC) and is known to be up-regulated in cells sensitive to ferroptosis [97]. In contrast, the silencing of the iron metabolism master regulator IREB2/IRP2 decreases sensitivity to ferroptosis [85].

### 3.2. Antioxidant Systems: The Last Defense before Ferroptosis Execution

The reduced glutathione (GSH)-dependent enzyme glutathione peroxidase 4 (GPX4), which is directly inhibited by RSL3, has emerged as the main endogenous inhibitor of ferroptosis due to its ability to limit lipid peroxidation by catalyzing the GSH-dependent reduction of lipid hydroperoxides to lipid alcohols. Indeed, genetics studies on *gpx4* knockout animals have probed the evidence of the role of GPX4 as the most downstream regulator of ferroptosis (for a review, please see [98]). If systemic *Gpx4* knockout mice display embryonic lethality [99], the tissue-specific conditional ablation of *Gpx4* will lead to different pathologic issues, including acute renal and hepatic injury, neurodegeneration, and defective immunity to infection, suggesting the role of GPX4 in development and tissue homeostasis. More importantly, ferroptosis-specific inhibitors (including Fer-1, Lip-1, and VitE) could prevent tissue damage, underlining the contribution of ferroptosis to several types of tissue demise. Furthermore, ferroptosis inhibitors could also prevent tissue damage in models of ischemia/reperfusion injury in the kidneys [100,101], liver [86,90], and heart [96], proving the pathophysiological relevance of ferroptosis. As its name indicates, GPX4 activity is affected by dysfunctions in cysteine metabolism, leading to the depletion of glutathione or GSH levels [102]. Cystine is imported into cells in exchange for glutamate by the X_c_^−^ system (SLC7A11(xCT)/SLC3A2 complex), then reduced in the cysteine required for the synthesis of GSH [103]. Among the ferroptotic inducers, erastin, sorafenib, sulfasalazine, and L-glutamate suppress cysteine transport [104]. Other pathways involved in the cysteine biosynthesis level and in the biosynthesis of GPX4 have been identified to regulate ferroptosis, including the trans-sulfuration pathway [105] and mevalonate pathway [106], respectively. More recently, connected with the mevalonate pathway, ferroptosis suppressor protein 1 (FSP1), previously known as apoptosis-inducing factor mitochondrial-associated 2 (AIFM2), was identified as a ferroptosis resistance factor acting independently of GPX4/glutathione activity [107]. By reducing CoQ10, a byproduct of the mevalonate pathway, to CoQ10-H2 with NADPH, FSP1 inhibits the propagation of PL peroxidation. Figure 2 illustrates the molecular interactions involved in the induction and regulation of ferroptosis.

## 4. Autophagy and Ferroptosis Regulation

### 4.1. Ferritinophagy: Drivers of Ferroptosis Initiation

The activation of ferritinophagy, a newly defined selective form of macroautophagy required for the specific lysosomal degradation of ferritin [108], the main cellular iron-storage protein, seems to occur during the early initiation stage of ferroptosis. Indeed, although it occurs to maintain the iron balance, recent discoveries have highlighted the importance of ferritinophagy with transferrin trafficking as critical determinants of ferroptosis sensitivity via an increase in the labile iron pool promoting ROS generation. As described above, ferritin is able to store up to 4500 iron atoms in a 24-subunit macromolecular complex formed by ferritin light and heavy chains (FTL and FTH, respectively), and its specific lysosomal degradation releases iron and supplies the cell’s iron need. However, excess iron fuels the Fenton reaction, leading to the production of highly toxic hydroxyl radicals. When ferritinophagy occurs, nuclear receptor coactivator 4 (NCOA4) was recently identified as a specific autophagic cargo receptor that selectively recognizes FTH via the C-terminal domain of NCOA4, binding a conserved surface arginine (R23) on FTH [109]. The first demonstration that ferroptosis is a selective autophagic cell death process was performed by Gao et al. using the ferroptosis inducers erastin and cystine starvation [110]. By using RNAi screening coupled with subsequent genetic analysis, they were able to identify multiple autophagy-related genes as positive regulators of ferroptosis. Consistently, the inhibition of ferritinophagy by the blockage of autophagy (with bafilomycin A1 and chloroquine) or knockdown of *NCOA4* (as well as *ATG3/5/7/13*) represses the accumulation of ferroptosis-associated cellular labile iron and lipid ROS and ultimately ferroptotic cell death [110,111]. These findings can be extended to other cancer cell lines (including human fibrosarcoma, human pancreatic carcinoma, leukemia, head and neck carcinoma) and to other ferroptotic inducers (including dihydroartemisinin (DHA)) [110,111,112,113]. In particular, DHA, a semi-synthetic derivative of artemisinin anti-malarial drug, accelerates ferritinophagy through the AMP-activated protein kinase (AMPK)/mammalian target of rapamycin (mTOR)/p70S6k signaling pathway to trigger ferroptosis in acute myeloid leukemia cells [112]. Notably, we demonstrated a newly identified mechanism in which the salinomycin/ironomycin-mediated iron sequestration in lysosomes promotes ferritin degradation and the Fenton reaction, leading to the toxic lipid ROS and ultimately and preferentially to ferroptosis in breast CSCs [39]. We described that a “vicious circle” occurs, leading to iron-dependent cell death, which we may call a “deathloop” (Figure 2). However, the role of autophagy remains to be clarified.

### 4.2. Autophagy: A Dual Role in Ferroptosis Execution

Interestingly, autophagy is able to selectively target other ferroptosis-related actors. Many ferroptotic inducers including erastin, RSL3, FIN56, and sulfasalazine promote the degradation of GPX4 protein which is the only enzyme capable of inhibiting lipid peroxidation by reducing phospholipid hydroperoxide [106,114]. Interestingly, GPX4 is targeted by HSP90/LAMP2A- and HSPA8/HSC70-mediated autophagy (also known as chaperon-mediated autophagy, CMA) and is thus involved in the execution of ferroptosis. However, HSPA5 increases GPX4 protein stabilization [114] supporting the role of HSPs in the regulation of ferroptosis. More importantly, this finding extended our knowledge of the role of other forms of autophagy in the regulation of ferroptosis [115]. In particular, lipophagy, the selective autophagic degradation of lipid droplets (LDs) that are complex spherical organelles stocking neutral lipids, leads to the release of free fatty acids and promotes lipid peroxidation in ferroptosis [116,117]. The knockdown of the LD cargo receptor *RAB-7A* (a member of the RAS oncogene family) or *ATG5* also limits lipid peroxidation-mediated ferroptosis [117,118]. In line with lipid metabolism, hypoxia inducible factor 1 subunit alpha (HIF1A), one of major transcriptional factors regulating hypoxic response, is able to regulate negatively RSL-3/FIN56-induced ferroptosis through the upregulation of fatty acid binding protein 3/7 (FABP3/7), which is involved in fatty acid uptake and lipid storage [119]. Interestingly, the SQSTM1/p62-dependent autophagic degradation of the key circadian clock protein/regulator ARNTL (newly defined as “clockophagy”) leads to the EGLN2/PHD1 (egl nine homolog 2/hypoxia-inducible factor prolyl hydroxylase 1)-mediated downregulation of *HIF1A*, thus promoting ferroptosis [119]. More recently, Song et al., demonstrated that AMPK-mediated Beclin-1 phosphorylation promotes erastin/SAS-induced ferroptosis by directly blocking system X_c_^−^/SLC7A11 activity and leading to GSH depletion [120]. Accordingly, the Beclin-1 activator peptide TAT-BECN1 and the RNA-binding protein ELAV-like RNA-binding protein 1 (ELAVL1) increasing *BECN1* mRNA stability also promote the SLC7A11/system X_c_^−^ inhibitor-mediated ferroptosis [120,121]. It is thought that HMGB1, more widely known as a danger signal in immune responses, positively regulates erastin-induced ferroptosis in leukemia cells through the mitogen-activated protein kinase (MAPK/JNK/p38)-mediated upregulation of TFRC expression [122]. Nevertheless, endogenous HMGB1 can presumably also act through the activation of autophagy, being a BECN1-binding protein or an activator of the BECN1/PIK3C3 complex involved in the induction of autophagosome formation [123,124]. Additionally, the intracellular iron exporter FPN is now identified as a substrate for autophagic elimination, and its degradation by SQSTM1/p62 promotes erastin/RSL3-mediated ferroptosis in vitro and in xenograft tumor mouse models [125]. Importantly, inducing the autophagic degradation of FPN overcomes ferroptosis resistance. In addition, the inhibition of mitochondrial iron accumulation by the upregulation of the mitochondrial iron exporter CDGSH iron sulfur domain 1 (CISD1, also termed mitoNEET) [126] and the increase in mitochondrial ferritin (FtMt) [127] inhibits ferroptosis suggesting that mitophagy, the selective autophagic degradation of mitochondria, can promote/accelerate ferroptotic cancer cell death. Nevertheless, by targeting the SQSTM1/p62-mediated degradation of KEAP-1, autophagy induces the stabilization of NFE2L2/NRF2 (nuclear factor (erythroid-derived 2)-like 2) protein and its transcriptional activity, which mainly mediates the antioxidant responses, thus preventing erastin and sorafenib-induced ferroptosis in hepatocellular carcinoma cells [128]. NFE2L2/NRF2 transcription factor actives the expression of a large number of genes encoding ferroptosis inhibitors including NQO1 (NAD[P]H quinone dehydrogenase 1), HMOX1 (heme oxygenase 1), FTH (ferritin heavy chain), GPX4, system X_c_^−^/SLC7A11, and lastly MT1 G (metallothionein 1G) [128,129,130,131,132]. Interestingly, the overexpression of GPX4, consistent with its anti-oxidant function, has been shown to inhibit ROS-mediated autophagy to prevent immunogenic cell death, another regulated-cell death modality [133]. Accordingly, some studies have recently revealed the interplay between mTORC1, the master negative regulator of autophagy, and GPX4 signaling, suggesting the existence of a feedback loop between autophagy and ferroptosis [134,135]. Altogether, these findings highlight the ability of some key proteins of autophagy to directly or indirectly act on key factors initiating or regulating ferroptosis, and thus positively or negatively modulate the sensitivity to ferroptosis.

## 5. Therapeutic Strategies to Target Ferroptosis in CSCs/Targeting Ferroptosis in CSCs

### 5.1. Through Manipulating Tumor-Suppressor p53

At present, accumulating evidence supports the importance of ferroptosis both in the suppression of tumorigenesis and in cancer therapies (for a review, please see [136]). The best examples are studies highlighting the role of the tumor suppressor p53 (also called the “guardian” of genome stability) in the maintenance of cell/organism integrity by regulating ferroptosis through its transcriptional activity or a transcription-independent mechanism [137,138]. On one hand, p53 promotes ferroptosis through the inhibition of system X_c_^−^/SLC7A11 expression [139] or the promotion of SAT1 (spermidine/spermine N1-acetyltransferase 1), thereby activating 15-LOX (arachidonate lipoxygenase) expression [140], or again through glutaminase 2 (GLS2) expression involved in glutaminolysis [141]. On the other hand, p53 suppress ferroptosis by sequestering directly dipeptidyl-peptidase 4 (DPP4) in the nucleus and inhibiting DPP4-dependent lipid peroxidation at the plasma membrane [142]. Interestingly, the expression of DPP4/CD26 defines a cancer cell subpopulation displaying CSC-like features and correlated with poor prognosis in human colorectal cancer [143,144], suggesting that this population could be effectively targeted by ferroptosis inducers. More recently, iPLA2b (a member of the calcium-independent phospholipase A2), newly identified as a p53-target gene, acts as a major suppressor of p53-driven ferroptosis in a GPX4-independent manner under ROS-induced stress in numerous cancer cell lines [145]. iPLA2b mediates the detoxification of phospholipids by releasing oxidized fatty acids and is overexpressed in many human cancers, including kidney renal clear cell carcinoma and acute myeloid leukemia [145]. Notably, the inhibition of endogenous iPLA2b promotes p53-dependent tumor suppression in xenograft mouse models [145]. This finding suggests that iPLA2b is a promising therapeutic target for activating ferroptosis-mediated tumor suppression without serious toxicity concerns. In addition of their role in lipid metabolism, the inhibition of phospholipase A2 could thus more efficiently trigger the cell death of CSCs [146]. Altogether, p53 exerts its effect in a highly context-dependent manner on the regulation of lipid peroxidation in ferroptosis. Nevertheless, p53 could also regulate ferroptotic cancer cell death via its action on the regulation of autophagy. Moreover, recent works have highlighted that Eprenetapopt (APR-246, PRIMA-1^MET^), known as a mutant-p53 reactivator, is also able to induce p53-independent ferroptosis by GSH depletion through its capacity to conjugate to free cysteine [147] in several human cancers, including AML [148] and esophageal cancer [149]. Interestingly, APR-246, which is already used in clinical trials [149], in combination with ferroptosis inducers have a synergistic anti-leukemic activity in vivo, opening new therapeutic opportunities in AML [148]. Notably, cysteine deprivation targets leukemia stem cells more efficiently, with no detectable effect on normal hematopoietic stem/progenitor cells [150]. Therefore, the induction of ferroptosis creates a new therapeutic strategy to target CSCs in different histological types of cancer, especially for drug-resistant tumors, with a low toxicity.

### 5.2. Through the Use of Ferroptosis Inducers Now Available

Indeed, through our recent work and other studies, iron homeostasis is now recognized as one of the hallmarks of CSCs in numerous/different human cancers, including breast cancer, ovarian cancer, prostate cancer, lung cancer, cholangiocarcinoma and glioblastoma. In this context, ferroptosis inducers (such as small molecules erastin and RSL-3/5, initially developed to selectively target tumor cells bearing oncogenic RAS and chemotherapeutic drug resistance) now existing as well as other yet-to-be-developed iron-driven cell death inducers, have therapeutic potential in anti-CSC therapy. In addition, many clinical drugs (including sorafenib, sulfasalazine, FIN56, FINO2, artesunate, and dihydroartemisinin) are even available for use in ferroptosis-mediated cancer therapies. However, they have not been tested in CSCs yet. Their actions are summarized in Table 2. We carried the first demonstration highlighting that the anti-CSC compound salinomycin and its synthetic analog ironomycin, target the cell death of breast CSCs more efficiently and specifically both in vitro and in vivo by ferroptosis in sequestering iron in the lysosome [39,40]. Furthermore, the inhibition of other negative regulators of ferroptosis, such as CD44 or newly identified heat shock protein b-1 (HSPB1), also has great potential in this anti-cancer therapeutic strategy. Indeed, the highly used CSC marker CD44 (specially CD44v isoforms) has been described to prevent ferroptosis by promoting GSH synthesis through the stabilization of the xCT/SLC7A11 subunit of the cystine importer system X_c_^−^ at the surface membrane in gastrointestinal cancer stem-like cells [151]. Therapy targeted to the CD44v-xCT system could thus impair the GSH-mediated ROS defense ability of CSCs and lead to the depletion of CSCs [152,153]. Heat-shock protein beta-1 (HSPB1) inhibits erastin-induced ferroptosis by affecting iron uptake and subsequent lipid peroxidation [154]. Interestingly, several studies have previously shown that HSPB1 is required for selective autophagy, including mitophagy and lipophagy [123,155,156]. In addition, linked to increased lipid metabolism by CSCs, targeting key players of fatty acids metabolism may prove a promising anti-CSC strategy to better trigger ferroptosis [146]. For example, the inhibition of fatty acid oxidation by Etomoxir impairs CSC self-renewal and tumorigenicity in a hepatocellular carcinoma (HCC) context and sensitizes HCC CSCs to sorafenib, which is a broadly used chemotherapy drug against HCC [157]. Compared to iron chelation strategies, the preferential iron loading in CSCs, such as through aminoferrocene-based therapies [158,159,160,161], may also be more therapeutically useful for enhancing their sensitivity to ferroptosis. More recently, Gao J. et al., performed an elegant demonstration of a new cancer therapy named gene-interfered-ferroptosis therapy (GIFT) by combining iron nanoparticles with cancer-specific gene interference, here targeting two iron metabolic genes (*FPN* and *LCN2*) both in vitro and in vivo [162]. Significant ferroptosis was induced in a wide variety of cancer cells, with only very little effect on normal cells. This cancer therapy based on gene interference-enhanced ferroptosis also resulted in a significant tumor growth inhibition and durable cure in mice, encouraging new efforts to be made in the study of ferroptosis and iron dysregulation in CSCs.

## 6. Conclusions and Perspectives

In summary, the accumulation of intracellular iron and iron addiction, inter alia, for their maintenance and expression of stem cells markers are newly identified hallmarks of CSCs. Iron therefore represents a vulnerability or Achilles’ heel of CSCs that could be therapeutically exploitable for more efficiently targeting cell death in anti-CSC therapies. On one hand, iron deprivation mediated by iron chelation strategies could initially constitute a first potential therapeutic approach. Indeed, iron chelators may interfere with some CSC-related markers/actors or signaling pathways and affect CSC expansion or ultimately induce cell death by apoptosis. However, this therapeutic strategy has shown its limits in hematological cancers even if the patients also suffer chronic iron overload leading to cardiac, hepatic, or endocrinal damage. In the case of other histological types of cancer, patients display more anemia, excluding the use of chelator molecules. On the other hand, manipulating iron accumulation via the induction of ferroptosis can constitute an effective strategy to target CSCs. Accordingly, CSCs rely on iron until they “die for” it. Indeed, CSCs have shown to be far more susceptible to ferroptosis than to apoptosis, thus initiating the development of new therapeutic perspectives. Furthermore, some ferroptosis chemical inducers are now available, including various FDA-approved drugs. In addition, triggering ferroptosis can synergize or enhance the anti-cancer capacity of conventional therapies, including chemotherapies and radiotherapies, by overcoming tumor resistance. Ferroptosis is also often described as autophagy-mediated cell death, indicating that the activation of autophagy may be involved in both the occurrence and regulation of ferroptosis. Indeed, autophagy (degradative pathways under its different forms) selectively targets various key actors of ferroptosis. However, autophagy also plays a dual role (promotor versus repressor) in ferroptosis that is dependent on many conditions. In particular, lipid metabolism is connected with both autophagy, through its impact on autophagosome membrane formation, and ferroptosis, through lipid peroxidation, which may play a critical role in the regulation of ferroptosis as part of a feedback loop. Thus, some molecular mechanisms in the regulation of ferroptosis remain largely unknown. In addition, our knowledge about the mechanisms of iron metabolism dysregulation, particularly in CSCs, must be improved in order to cure cancer with a low toxicity and develop personalized cancer therapies targeting CSCs through ferroptosis.

## Figures and Tables

**Figure 1 cells-10-02981-f001:**
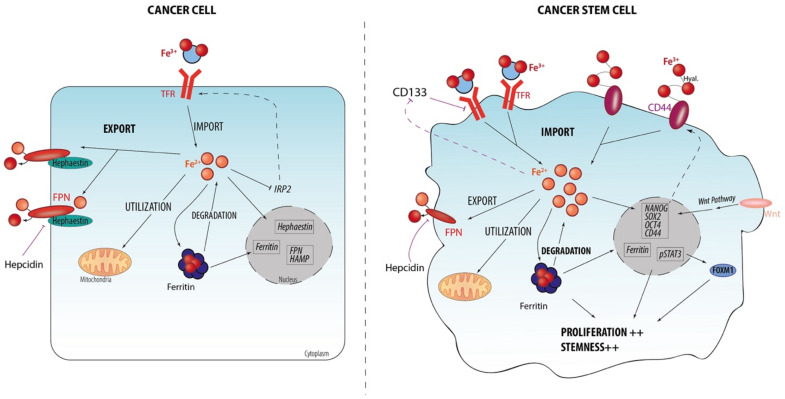
Increased iron metabolism drives CSC expansion and maintenance (compared to non-CSC counterparts). Expression of key proteins involved in iron trafficking are differentially expressed between cancer cells and cancer stem cells. Cancer cells or non-CSCs: low levels of TFR expression (regulated by IRP2, which is itself regulated by the iron status of the cells to maintain iron homeostasis) in charge of iron uptake, and high levels of FPN (regulated by hepcidin, iron master regulator) and/or hephaestin (ferroxidase) expression involved in iron export, collectively lead to low level of intracellular iron. CSCs: High level of CD44 (stem cell marker) via its interaction with iron-bound hyaluronates and TFR via Tf/2Fe^3+^ increase iron uptake, whereas a downregulated FPN level decreases iron efflux. This leads to a higher intracellular iron level, directly supporting the expression of some stem markers (including CD44 or CD133 regulating the endocytosis of TFR/Tf), the Wnt pathway (activator of *CD44*, *SOX2*, *NANOG*, and *OCT4* expression), and STAT3-FOXM1 signaling. Thus, this increasing iron metabolism leads to CSC proliferation and supports CSC/stemness behavior. Arrowhead or stop lines indicate promotion/activation or inhibition, respectively.

**Figure 2 cells-10-02981-f002:**
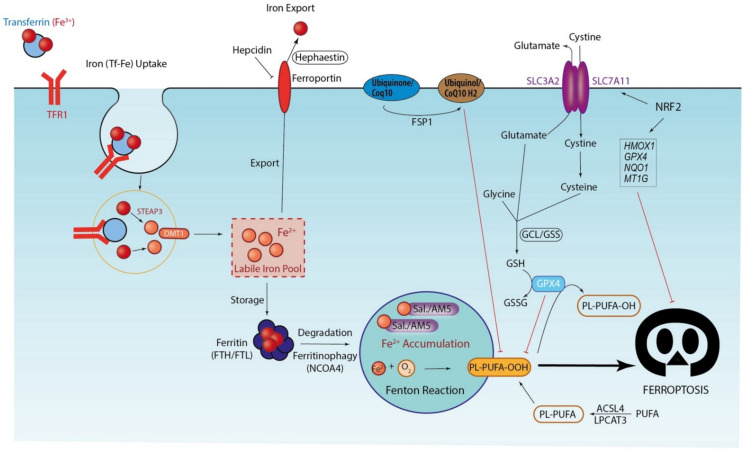
Molecular interactions involved in the induction and regulation of ferroptosis. Decisive characteristics of ferroptosis resulting from phospholipid peroxidation (LOOH) are: (1) its dependence on iron, under the Fenton reaction (Fe^2+^ + H_2_O_2_ → Fe^3+^ + OH + OH^−^), which can be caused by: (i) the increased expression of key players for iron uptake, including iron-binding transferrin receptors (TfR1/TFRC), or even DMT1 or STEAP3 that directly bind iron; (ii) the degradation of ferritin (FTH/FTL complex), which is the iron-storage protein, involving ferritinophagy, an NCOA4-dependent autophagic process to release iron in the lysosome; (iii) lysosomal iron sequestration, as induced by Sal./AM5; and iv) the hepcidin-mediated inhibition of ferroportin expression, which is involved by the export of iron and thus promotes iron retention. (2) Disturbances in the GSH/GPX4 axis that ensures the redox balance; or in the same order, disturbances in the level of CoQ10 (a byproduct of the mevalonate pathway) with its FSP-1 reductase that were recently found to prevent the peroxidation of membrane lipids. ACSL4 and LPCAT3, which are involved in the esterification of PUFA in membrane phospholipids (PL-PUFA), are key actors in sensitivity to ferroptosis. NRF2, the master antioxidant regulator, is also able to prevent ferroptosis by activating the expression of a large number of genes encoding ferroptosis inhibitors including NQO1, HMOX1, FTH, GPX4, system X_c_^−^/SLC7A11, the GCL/GSS enzymes involved in the GSH synthesis, and lastly MT1 G. Arrowhead or stop lines indicate promotion/activation or inhibition, respectively.

**Table 1 cells-10-02981-t001:** CSC-related markers in different cancers. Some of them (indicated in bold) are connected with iron homeostasis.

Markers	Cancer	Ref
**CD44+**CD24-	Breast	[10,11]
ALDH1+	Colon, Brain, Acute Myeloid Leukemia, Breast, Stomach, Melanoma	[12,13,14,15,16,17]
**CD133+**	Brain, Colon, Pancreas, Lung, Ovarian, Prostate, Stomach	[18,19]
**CD44+**ALDH1+	Ovary	[10,20]
**CD44+**a2b1^high^**CD133+**	Prostate	[10,18,21]
ABCB5	Melanoma	[22]
**CD44+**	Colon, Head and Neck	[10,23,24]
CD24+	Colon	[25]
CD166+	Colon, non-small cell lung cancer	[26,27,28]
**CD133+**EpCAM+	Liver	[18,29]
**CD44+**EpCAM+	Colon	[10,23]
ESA+**CD44+**CD24+	Pancreas	[30]
CBX3+ABCA5+	Osteosarcoma	[31]
LGR5+	Colon	[32]
CD90+	Liver	[33]
CD34+CD38-	Acute Myeloid Leukemia	[34]
CD34+CD38+CD19+/CD34+CD38-CD19+	Leukemia	[35]

ABC, ATP-binding cassette; ALDH1, Aldehyde dehydrogenase 1; CBX3, Chromobox homolog 3; EpCAM, Epithelial cell adhesion molecule; ESA, Epithelial specific antigen; LGR5, Leucine Rich repeated-containing G-protein coupled receptor 5.

**Table 2 cells-10-02981-t002:** Ferroptosis inducers in different cancers. Most ferroptosis inducers are classified into 2 categories on the basis of their mechanism of action: Class 1 inducer: inhibition of system Xc^−^ leading to GSH depletion; Class 2 inducer: inhibition of GPX4 activity leading to lipid peroxidation.

Name	Class/Action	Cancer	Ref
Erastin and its derivatives (better stability): Aldehyde erastin Piperazine Morpholine erastin II	Targets the mitochondrial voltage-dependent anion channel 2/3 (VDAC2/3); Class I inducer through the binding of SLC7A5, a subunit of system Xc^−^/induces of Beclin1-SCL7A11 complex formation/inhibits cystine uptake leading to GSH depletion/induces also the GPX4 protein degradation	*Kras*-mutant tumor cells, lung, leukemia, CRC	[73,163,118]
Sulfasalazine (SAS)	Class I inducer/Induces Beclin1-SCL7A11 complex formation/inhibits cystine uptake leading to GSH depletion	Lymphoma, SCLC, prostate cancer, breast cancer, glioblastoma, combined with dyclonine, targets ALDH3A1^+^ tumors cells in head and neck squamous cell carcinoma and in gastric tumors, leukemia, pancreatic cancer	[85,163,164,165,166,167,168]
Sorafenib	Class I inducer/inhibits the activity of system Xc^−^	Liver, kidney, lung or pancreatic derived cell lines, AML, HCC, neuroblastoma, NSCLC, RCC	[95,169]
Tat-beclin1	Enhances erastin anti-cancer activity/direct inhibitor of the activity of system Xc^−^/leads to lipid peroxidation	Colon, pancreas, lung (NSCLC), cervical	[120]
Lanperisone (FDA-approved drug)	Class I inducer/inhibits cystine uptake leading to GSH depletion	*Kras*-mutant tumor cells	[170]
RSL3/5	Class 2 inducer/binds GPX4 to inhibit its enzymatic activity/induces also the GPX4 protein degradation	*Kras*-mutant tumor cells; AML cells, Head and neck cancer, Colorectal cancer	[97,106,113,171,172,173]
FIN(*ferroptosis inducing*)56	Class2 inducer, downregulates GPX4 expression at mRNA level/targets GPX4 degradation/causes depletion of mevalonate-derived coenzyme Q10 (CoQ10)	Osteosarcoma, lung adenocarcinoma, fibrosarcoma	[106]
FINO2 (endoperoxide-containing 1,2-dioxolane)	Class 2 inducer/represses indirectly the enzymatic function of GPX4 leading widespread lipid peroxidation/is able to oxidize ferrous iron directly/can also oxidize lipids, providing another source of lipid peroxides	NIH60, a range of cancer cell lines from different tissues, engineered cancer cells such as RCC cells and fibrosarcoma cells	[174,175]
Artesunate (artemisinin derivative, anti-malaria drug)	Targets iron/induces ferritin degradation leading to the lysosomal iron release and Fenton reaction with ROS	mutationaly-active *Kras* pancreatic ductal adenocarcinoma cell lines, lymphoma	[176,177,178,179]
Dihydroartemisinin (DHA, semi-synthetic artemisinin derivative)	Targets iron/inducs the autophagy-dependent degradation of ferritin by regulating the activity of the AMPK/mTOR/p70S6K pathway	Leukemia, glioma, head and neck cancer;	[112,180]
BSO (Buthionine sulfoximine)	Targets GCLC; prevents GSH synthesis;	HCC, *Kras*-mutant tumor cells SCLC cancer	[128,172,181]
Siramesine and lapatinib	Targets iron, decreases the expression of FPN and Ferritin and increases iron uptake through upregulation of TRFC;	Breast cancer cells	[182,183]
Salinomycine and its synthetic derivate Ironomycin	Sequesters lysosomal iron leading to cytoplasmic iron depletion/increases iron uptake through the up-expression of IRP2 and TFRC, along with the accelerated lysosomal degradation of ferritin	Breast cancer stem cells, ovarian cancer	[39,40,184]
Cyst(e)inase	Targets enzymatic degradation of cystine/cysteine, restricting its availability to cancer cells and triggering ferroptosis	Breast cancer, chronic lymphocytic leukemia, melanoma, pancreatic cancer, prostate cancer	[185,186,187]
Statins (Fluvastatin, Pravastin, lovastatin and simvastatin), inhibitors of HMGCR (HMG-CoA reductase), a rate-limiting enzyme in the mevalonate pathway	Are able to inhibit the biosynthesis of selenoproteins including GPX4 and CoQ10	Breast cancer, HCC, AML, MM;	[188,189,190,191]
Eprenetapopt (APR-017/PRIMA-1, APR-246/PRIMA-1^MET^)	Mutant-*p53* reactivators, has also the ability to conjugate free cysteine leading to GSH depletion and interferes with iron-sulfur cluster biogenesis	AML	[148]
Ferumoxytol (Feraheme, FDA-approved iron oxide nanoparticle)	Targets iron, fuels Fenton reaction leading to harmful production of ROS;	Leukemia cells	[160]
Iron salophen complexes (chemicaly-engineered transition-metal complexes)	Generate lipid ROS leading to ferroptosis	Leukemia, neuroblastoma cell lines	[161]
Fenugreek (trigonelline)	Inhibits NRF2 leading the blockage of MT-1G expression, and GSH depletion	HCC cells, head and neck cancer cells	[128]

AML, Acute Myeloid Leukemia; CRC, Colorectal cancer; HCC, Hepatocellular Carcinoma; MM, Multiple Myeloma; (N)SCLC, (Non)-Small-Cell Lung Cancer; RCC, Renal Cell Cancer.

## Data Availability

Not applicable.

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
