# Peer review of "Ferroptosis: Cancer Stem Cells Rely on Iron until “to Die for” It"

_cells, 2021, doi:10.3390/cells10112981_

Round 1

Reviewer 1 Report

The work is an extensive review of the literature about the alterations of iron homeostasis in cancer stem cells. In the initial part, they mainly review the works before the description of ferroptosis, and ferroptosis is the focus of the second part. The work is complex and sometimes difficult to follow. And some sentences need to be rephrased, and English should improve.

- The iron addiction or iron excess in the CSCs is described throughout the work, but the initial part includes various studies that show that iron deprivation is a potential therapeutic approach, while in the ferroptosis part the studies suggest the indiction of iron accumulation as a therapeutic approach. Thus, the authors should discuss this contradiction in the conclusions.

- table 1 should include the references to the papers that mention the markers

- Some molecule abbreviations or codes are not explained, examples are FEROSTEM (page 3, Line 63), OMS (L 117), FOXM1 (L 124). In addition, they use both FTH and FTH1, and FPN and FPN1, which may create confusion.

- L. 81, “the first demonstration…..” the sentence is missing something and should be rephrased.

- similarly at L 157, “JARID1B identify …” what is JARID1B?

- L 244 “excessive iron feels Fenton’s reaction….” correct.

- L 260 “vicious cercle” corrected

- L 315 “could be targeting….” correct

- the conclusions are too vague and general, some more focus would help.

Reviewer 2 Report

The manuscript by Emma Cosialls et al is a well-written and also updated review about the key role of iron in cancer stem cell pool expansion and progression. The authors summarize different facets of the iron involvement in cancer stem cells expansion, focusing also on the role of iron metabolism-related proteins.

Considering that, among the iron-related proteins, FTH represents a key hub also in ferroptosis, I believe that the authors should better dissect the role of fTH in CSCs expansion.

A survey of the literature show that FTH regulated cancer stem cells expansions by acting either on the expression of CSCs markers, EMT process and spheroids formation in many solid tumor types (Chirillo R et al, Frontiers in Oncology, 10.3389/fonc.2020.00698), (Salatino A et al, Oxidative Medicine and Cellular Longevity, 10.1155/2019/3461251)

Furthermore, it has been demonstrated the role of FTH in cancer cell differentiation (Zolea F et al, IJMS, 10.3390/ijms18102167).

The authors should also pay attention to minor english typing errors.

Reviewer 3 Report

This review paper by Cosialls et al is to summarize the crosstalk between cancer stem cells and iron/ ferroptosis.    Concerns 1. The authors are suggested to make a table or figure, which can show the differences of iron demand/metabolism in tumor cells and cancer stem cells instead of word description in the section of iron and CSC.    2. In Table 1, the authors are suggested to add a brief description and mechanism about the association between CSC-related markers and iron metabolism.    3. The authors are strongly encouraged to draw a figure to present the connection or mechanism that iron metabolism contributes to the growth of cancer stem cells and tumorigenesis.   4. Some typing in Figure 1 is not very clear. Please re-type it.    5. A spelling errors in line 172, Ferrostatin-1. 

Round 2

Reviewer 1 Report

the answers to the points I raised are satisfactory and the manuscript improved

Author Response

Thank you again for your positive comments to improve the quality of our manuscript. Please find the revised manuscript.

Reviewer 2 Report

I am satisfied by the new version of the manuscript and by the corrections made by the authors.

I suggest to accept in the current form.

Author Response

(The authors gave the same response as above.)

Reviewer 3 Report

1. A spelling error of FSP1 (Ferroptosis suppressor protein 1) in Figure 2. 

2. Please unify the Nrf2 or NRF2.  

Author Response

Thank you very much for your careful reading. We did the corrections.

1. The spelling error of FSP1 in Figure 2 was corrected.

2. We unified it by ‘NRF2’.

Thank you again for your positive comments to improve the quality of our manuscript. Please find the revised manuscript.